# Serum Albumin Kinetics in Major Ovarian, Gastrointestinal, and Cervico Facial Cancer Surgery

**DOI:** 10.3390/ijerph19063394

**Published:** 2022-03-14

**Authors:** Cyrus Motamed, Lucie Mariani, Stéphanie Suria, Gregoire Weil

**Affiliations:** 1Department of Anesthesia, Institut de Cancerologie Gustave Roussy, 94080 Paris, France; stephanie.suria@gustaveroussy.fr; 2Department of Anesthesia and Intensive Care, APHP Hopital Pitié Salpétrière, 75013 Paris, France; lucie.mariani@aphp.fr; 3Anesthesia Department, Centre Hospitalier d’Orleans, 45100 Orléans, France; gregoire.weil@chr-orleans.fr

**Keywords:** major cancer surgery, serum albumin kinetics, albumin

## Abstract

Hypoalbuminemia in major cancer surgery can lead to postoperative short and long-term complications. Our study was designed to detect albumin variations in three major cancer surgeries: ovarian debulking (DBK), major abdominal gastrointestinal surgery (ABD), and major cervico-facial, or ear, nose and throat cancer surgery (ENT). Single-center prospective study inclusion criteria were non-emergency procedures scheduled to last at least five hours. We performed hourly perioperative monitoring of the patients’ albuminemia and hemoglobinemia. Electronic charts were followed for at least five years for survival analysis. Sixty-three patients were analyzed: 30 in the DBK group, 13 in the ABD group, and 20 in the ENT group. There was a significant difference in albumin decrease between the ENT group and the two others (−19% at six hours in the ENT group versus -49% in the debulking group and −31% in the ABD group (*p* < 0.05). There was no significant difference between the DBK and ABD groups. The decrease in hemoglobin was not significantly different between the groups, and no significant difference was observed in long-term survival. DBK and ABD surgery yielded significant hypoalbuminemia. Therefore, the extent of decrease in serum albumin is probably not the only etiology of the specific postoperative complications of these major surgeries. No significant difference was noticed in five-year mortality, and no correlation was found in relation to the degree of intraoperative albumin kinetics.

## 1. Introduction

Albumin constitutes a major percentage of total human plasma. The three main functions of serum albumin are osmosis, transport, and nutrition. Many studies have revealed positive correlations between perioperative serum ALB and outcomes. For example, preoperative hypoalbuminemia is reported as a predictor of incision site infections [1], survival [2], or mortality [3,4]. Ryan et al. [5] described the association between postoperative day (POD) 1 hypoalbuminemia and postoperative complications after esophagectomy. Sang et al. [6] have also reported that hypoalbuminemia on POD 2 was significantly related to acute kidney after liver transplantations. Serum ALB also decreases because of trauma and increased capillary leak [7]. Norberg et al. [8] reported a steep decrease in serum ALB levels down to 33% after major abdominal surgery while CRP levels were not yet significantly affected.

Decreases of 14 and 19% in cut-off values for ALB have been suggested for recognizing patients at a high risk for complications after gastric cancer resection [9,10]. The role of albumin as an independent prognostic marker in ovarian tumors is debated [11,12]. The decrease in plasma albumin concentration following surgery is well known and documented [13], with these authors finding it to be a significant predictor for clinical outcomes in non-cancer surgery; however, the temporal pattern and the effect of different types of surgery are not well assessed or understood.

The primary objective of this observational study was to assess the changes in intraoperative albumin following three major cancer surgeries, while the secondary objective was to assess mid and long-term follow-up and possible parameters affecting survival.

## 2. Materials and Methods

We conducted a prospective, single-center, observational study in a cancer hospital between December 2011 and May 2012. Our work was submitted to the Paris XII Ethical Protection Committee (reference SC11-022, accepted on 21 November 2011). Informed consent was obtained from all patients after obtaining oral information from the investigational procedure.

### 2.1. Inclusion Criteria

Patients requiring debulking (DBK) ovarian surgery, major abdominal surgery (ABD), or major cervicofacial surgery (ENT) with a duration of more than five hours were included in order to have a minimum of four data points of assessment and justify the placement of an arterial catheter as part of the local hemodynamic monitoring protocol.

### 2.2. Non-Inclusion and Exclusion Criteria

The exclusion criteria were age under 18, laparoscopic surgery, intraperitoneal gelatinous tumors, surgery with intraperitoneal chemotherapy, emergencies, and an expected intervention duration of less than five hours. Other exclusion criteria were failure of arterial catheter placement and actual surgery duration of less than five hours. The transfusion of blood products in the first five hours of the operation resulted in termination of the protocol; however, the values already collected were kept.

For ovarian and major abdominal surgery, patient-controlled epidural analgesia has been systematically proposed for postoperative analgesia in the absence of contraindications or refusal of the patients. A catheter was inserted under local anesthesia before the induction of general anesthesia in T9–T10 spaces. Epidural analgesia was not used during the surgeries. Intravenous anesthesia was induced with remifentanil (target-controlled infusion 3 ng/mL), propofol (2–3 mg/kg) and atracurium (0.5 mg/kg) or succinylcholine in case of rapid sequence intubation. Anesthesia was maintained with desflurane for a bispectral index between 40 and 60. Standard monitoring included electrocardiogram, capnography, airway pressure, pulse oximetry, neuromuscular monitoring of the adductor pollicis, radial, or femoral arterial blood pressure, esophageal temperature, and urinary output. A warming device was connected to a warming blanket. Patients had intravenous antibiotic prophylaxis before the incision and regularly during the surgery, according to the local protocol for each type of surgery.

We routinely applied a protocol for fluid management to avoid intraoperative hypovolemia or fluid overload based on goal-directed therapy with individual maximization of flow-related hemodynamic parameters. For this surgery, we systematically monitored arterial pulse pressure variations to guide fluid management. Blood and fluid loss (aspiration, weight of the compress, ascites), and urinary output were strictly monitored. The perfusion was based on 5 mL/kg/h of crystalloid, and we administered hydroxyethyl starch (250 mL) following the loss of exudates or blood when the hemodynamic parameters were in favor of hypovolemia (pulse pressure variation >13%). In our hemodynamic protocol, arterial hypotension was defined by a mean arterial pressure less than 65 mm/Hg or a decrease of 20% of the basal systolic arterial pressure in the case of arterial hypertension. Hypotension was treated with a bolus of ephedrine or phenylephrine, or fluid challenge if necessary. Norepinephrine infusion was used in cases of prolonged arterial hypotension of more than 10 min.

If necessary, packed red blood cells were administered to obtain a hematocrit upper to 25% for both abdominal surgeries, but for free flap cervicofacial surgeries, a threshold of 30% (hemoglobin = 10 g/dL), fresh frozen plasma was administered for a prothrombin ratio > 60% and platelets for a rate upper to 100,000/mm^3^. Supplementary measurements were performed if necessary.

Intraoperative pain management consisted of ketamine (0.3 mg/kg) at the incision, followed by an infusion of 0.15 mg/kg/h, which was stopped 1 h before the end of the surgery. Epidural analgesia for abdominal surgery was not used to avoid a deep sympathetic blockade in cases of major hemodynamic variations or acute hemorrhage. Sixty minutes before the end of the surgery, an epidural bolus of 0.2% ropivacaine (20 mL) and sufentanil (10 µg) was injected. In cases of contraindication or patient refusal of the epidural, morphine (0.15 mg/kg) was administered 1 h before the end of the surgery. Postoperative analgesia was achieved with a patient-controlled analgesia regimen using either ropivacaine 0.2% for epidural with low dose morphine or intravenous morphine alone.

Following surgery, patients were transferred to the postanesthetic care unit (PACU) and then to postoperative surgical intensive care.

The demographic characteristics of the patients were noted: age, weight, height, type intervention, nutritional status (undernutrition if weight loss >10% over the last six months and/or if albuminemia < 35 g/L), and type of surgery. Preoperative hemoglobin and albuminemia values were collected. Transfusion with blood-derived products in the first six hours of the operation resulted in termination of the protocol; however, the values already collected were retained.

Heart rate, blood pressure, and SpO2 were collected continuously. Cardiac index and stroke volume variation (SVV) were calculated continuously by analysis of the pulse wave contour (Vigileo^®^ monitor version 3.01-Edwards Lifesciences, Irvine, CA, USA). Hourly diuresis and the volumes and types of fluids administered were recorded.

The losses corresponding to evaporation from the operating site and basic metabolic needs had to be compensated for by crystalloids up to 5 mL/kg/h for the DBK and ABD groups versus 2 mL/kg/h during ENT surgeries. An hourly input/output balance associated with weighing the compresses allowed iso-volume compensation of blood loss by colloids. In the event of hypovolemia, defined by a drop in the cardiac index (10% of the initial value after induction over 60 min) and an increase in the SVV (>15%), a rapid loading test with 250 mL of colloid was performed. Blood samples were taken in the order defined in Figure 1 at the incision time through six postoperative hours. The collection of biological data was interrupted in the event of transfusion of packed red blood cells, fresh frozen plasma, or platelets.

### 2.3. Endpoints

The primary endpoint was the comparison of the overall intraoperative variation of albumin in these three types of surgery variations in albuminemia (expressed as % loss compared with the preoperative value) between the DBK group and the ABD group. The secondary outcome was overall survival at five years following the search for possible factors influencing survival.

The comparison between both abdominal surgeries and major ENT surgery for the decrease in albuminemia (expressed in % loss compared to the preoperative value) and for the decrease in hemoglobinemia (expressed in % loss compared to the preoperative value) was performed to take intraoperative bleeding into account.

### 2.4. Statistical Analysis

The characteristics of the patients were expressed as mean (SD) for continuous variables and as frequencies (%) for categorical variables. As the number of patients was not equal between the surgery groups and the duration of surgeries was not the same, a two-way ANOVA could not be used. We compared the groups using a one-way ANOVA of the area under the curve (AUC) of the intraoperative variation of albumin and hemoglobin.

The secondary endpoint was overall survival at five years. Patient characteristics and endpoints were compared between the patients according to their surgery group using the Student’s *t*-test or χ^2^ test. Survival according to surgery was compared with a Kaplan–Meier curve. The characteristics and intraoperative variables were then compared according to survival status. Finally, a Cox proportional hazards model was constructed, taking into account intraoperative albumin variation and the significant variables affecting the outcome.

Two-tailed *p*-values of <0.05 were considered significant. All analyses were conducted using R 4.1.2 for Windows (R Foundation for Statistical Computing, Vienna, Austria). The sample size was inspired by a previous investigation with a minimum of 10 major abdominal surgery patients in whom a 40% decrease in serum albumin was observed in the perioperative period [8].

## 3. Results

Seventy-three patients were included in the study between December 2011 and May 2012. Ten patients were excluded from final analysis (Figure 1, flowchart). Therefore, the DBK group consisted of 30 patients, the ENT group consisted of 20 patients, and the ABD group consisted of 13 patients. Demographic characteristics are displayed in Table 1. The groups differed in age and weight. Major ovarian surgery consisted of 16 optimal and 14 suboptimal debulking, all these patients had prior (4–6) cycles of chemotherapy and (3–4) weeks intervals before surgery. Major abdominal surgery included 5 major colorectal surgeries, 2 gastrectomy, 2 hepatectomy, 2 pancreatico duodenoctomy, 1 Lewis Santy, and 1 retroperitoneal sarcoma. All twenty cervico-facial surgeries included total tumor removal with free flap reconstruction.

All limits of excisions were free of disease in all three sub-groups of patients.

The fluid administration in the ENT group was significantly lower than in the other two groups, whether for crystalloids, colloids, or total volumes of solutes infused (*p* < 0.001 for all tests). Patients in the DBK group received 14 (11–17) mL/kg/h of crystalloids and 2.2 (1.54–3.99) mL/kg/h of colloids intraoperatively. Patients in the ABD group received 8 (8–15) mL/kg/h and 1.35 (0.79–2.41) mL/kg/h respectively versus 8 (5–10) mL/kg/h and 0.5 (0.00–0.95) mL/kg/h for the ENT group (Table 2). Regression model for the maximum variation of albumin with regard to ASA score and DBK model are displayed in (Table 3). Both ENT and ABD group were significantly different in comparison to the DBK group.

The initial albuminemia was 32.2 ± 4.2 g/L for the DBK group, 31.5 ± 5.2 g/L for the ABD group, and 32.9 ± 4 g/L for ENT (NS). The AUC variations in boxplots for albumin and hemoglobin are shown in Figure 2 and Figure 3. There was a significant difference between the kinetics of the albuminemia of the DBK/ABD groups and that of the ENT group (AUC of 1.3 [0.85–1.8] and 1.1 [0.54–1.2] vs. 0.43 [0.26–0.76], respectively, *p* < 0.001).

The percentage of variation of serum albumin is represented in boxplots, the decrease was significantly less important in ABD and ENT group overall.

The AUC comparison detect a significant difference between ENT and DBK group.

The AUCs of intraoperative hemoglobin loss did not differ between the three groups, with an AUC of 0.05 [0–0.27] in the DBK group, 0.03 [0–0.28] in the ABD group, and 0.08 [0–0.15] in the ENT group (Figure 4).

The correlation between albumin and hemoglobin loss is illustrated in Figure 5. Albuminemia showed a decrease of approximately 20% at H + 6 in the ENT group, 40% in the ABD group, and 45% in the ovarian surgery group.

No significant correlation was found between delta HB and delta Alb and the total volume infused. Correlation coefficient 0.18–0.26.

The demographic characteristics of the patients with regard to outcome considering BMI, age, ASA score, malnutrition status, cardiovascular disease, smoking habit, preoperative serum albumin, creatinine clearance, and types of surgery were not different between survivors and non survivors. The Kaplan–Meier analysis with regard to the outcome at five years is shown in (Figure 6). No significant difference or correlation was noticed with regard to the outcome or the decrease in albumin. The Cox regression hazard ratio regression for survival analysis did not detect any significant factors in the five-year survival analysis (Figure 7).

## 4. Discussion

This study showed that a decrease in serum albumin occurs in major ovarian and abdominal surgeries, as well as major ENT surgeries. We also found a significant correlation coefficient between the variations in albuminemia and hemoglobinemia, particularly in the major ENT group. The strong correlation we found in this specific group of patients suggests that the variations in albuminemia are largely explained by the blood loss and dilution necessary to maintain normovolemia. This result can, in part, be applied to major abdominal surgeries. The coefficient of correlation is weaker but still significant. For the DBK group, the lower correlation coefficient suggests that this mechanism plays a minor part, which is supported by other studies [14,15].

Hypotonic colloids have been implicated in the mechanism of the extravasation of albumin. In this case, there would be an effect on the quality of the fluid rather than the amount administered; this effect is identical for non-hypotonic colloids. It should be noted that colloids also yield a decrease in hepatic albumin synthesis [16]. However, these mechanisms do not seem to occur over a period of time as short as that of standard major surgery [17,18,19]. To our knowledge most of the studies in this field considered one type of surgery only, mostly major abdominal surgeries but our study also included major cervicofacial cancer surgery with free flap reconstruction.

Changes in albumin kinetics have been explored in previous investigations during major abdominal pancreatic surgery in a similar number of patients; however, these studies have not assessed intraoperative kinetics, although no significant difference was noticed before and 2 days after surgery [13,20]. Early perioperative decrease in serum albumin levels in patients undergoing major abdominal, urological, or thoracic surgery have been reported to predict adverse postoperative outcomes [21], such as increased 30-day mortality [4] but can also predict survival or length of stay [13].

The mechanisms of early postoperative albumin decrease combined altered metabolism, blood loss/dilution, and redistribution into the third space due to capillary leakage [7], which itself is related to the magnitude of the systemic inflammatory response. The intensity of the decrease in serum albumin has been previously reported to be between 11 and 14% [21] and is reported to affect outcomes. In this study, the mean decrease was above this value for all three types of major cancer surgeries. With regard to major cancer ENT surgery, our data are in accordance with a previous study on complex traumatic limb injury patients and hypoalbuminemia in patients having microsurgical flaps for treatment of their injuries. This did not influence the occurrence of complications that required surgical re-intervention; however, it was associated with prolonged hospital stay [22]. Another important issue is the DBK group which obviously considered only female patients which could influence the overall albumin concentrations [23], but since we considered only the percentage of decrease, we believe this would not affect our results and conclusions.

This study has some shortcomings. The sample size was small after final exclusions, and our group of abdominal surgeries did not have enough patients, mainly because many surgeries in the study period lasted less than five hours. This contributed partly to the uneven distribution.

Our short and long-term follow-up did not reveal any modified outcomes with regard to decreased serum albumin, type of surgery, age, malnutrition, or fluid administration, confirming the heterogeneity of factors in overall cancer mortality. Furthermore, affecting outcome results of this specific subgroup of patients was the diversity of tumors and surgery, indeed 48% of surgeries were considered suboptimal however surgeons in our specialized cancer hospital perform aggressive surgery targeting maximum disease removal and the difference between optimal and suboptimal surgery remained slim.

In addition, we did not consider the amount of serum albumin at the time of primary diagnosis of the disease since it was not the primary objective of the study; nevertheless, albumin is basically an additional tool for surgery risk stratification rather than a nutritional marker. Finally, our mortality assessment was only a retrospective or passive follow-up, which is less accurate than active follow-up combined with cancer registry data [24].

## 5. Conclusions

The decrease in serum ALB was present in all three types of major cancer surgeries but was less pronounced in major cervicofacial surgeries. We could not detect in our small series of patients a significant relation between the extent of the serum ALB decrease and short- or long-term postoperative outcomes and survival, further studies with larger sample sizes are necessary to adequately investigate oncologic outcomes.

## Figures and Tables

**Figure 1 ijerph-19-03394-f001:**
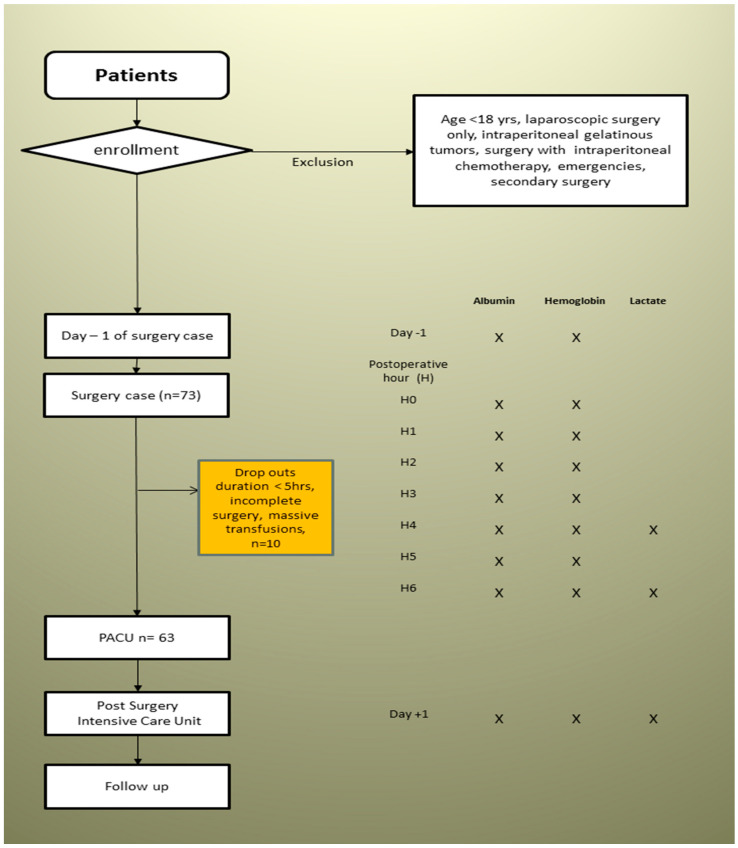
Flowchart.

**Figure 2 ijerph-19-03394-f002:**
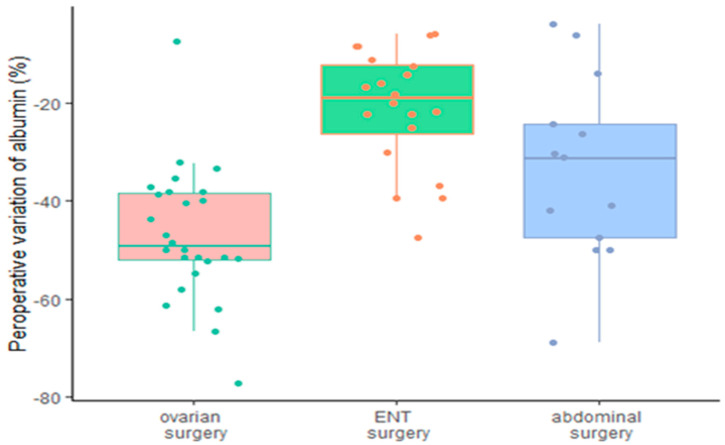
Maximum intraoperative albumin variation.

**Figure 3 ijerph-19-03394-f003:**
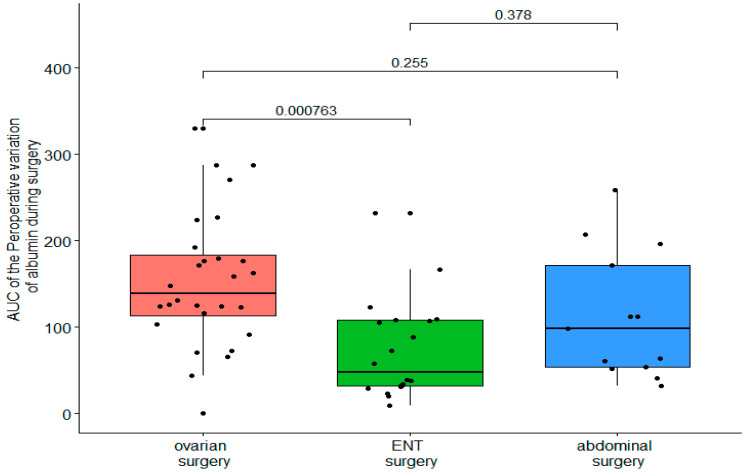
Comparison of the AUC of intraoperative variation of albumin between surgery groups.

**Figure 4 ijerph-19-03394-f004:**
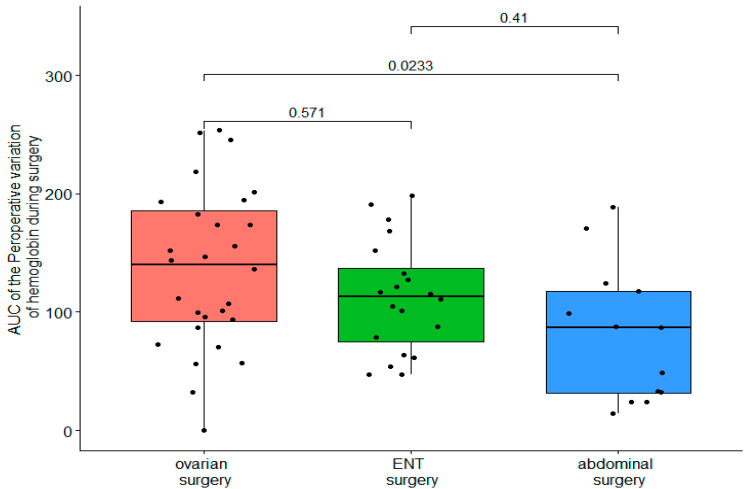
Comparison of the AUC of intraoperative variation of hemoglobin between surgery groups.

**Figure 5 ijerph-19-03394-f005:**
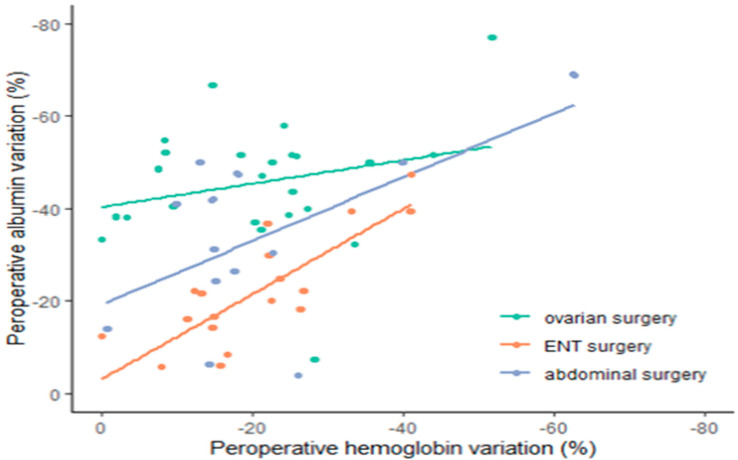
Correlation of albumin and hemoglobin variations.

**Figure 6 ijerph-19-03394-f006:**
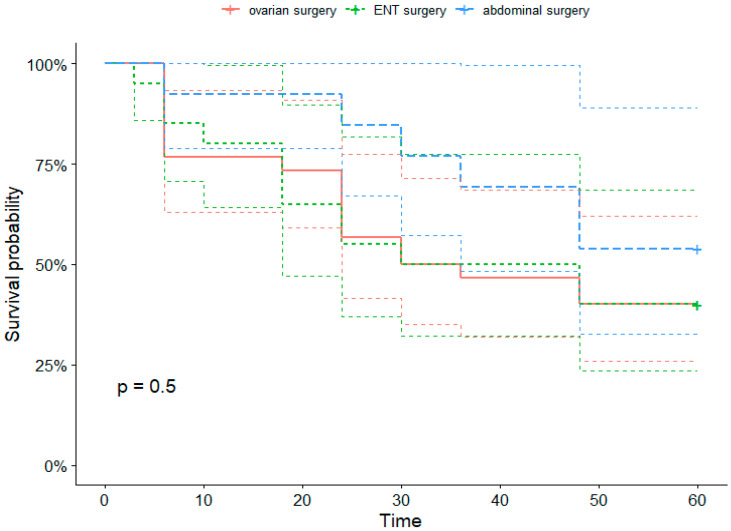
Kaplan–Meier representation of five-year survival according to type of surgery.

**Figure 7 ijerph-19-03394-f007:**
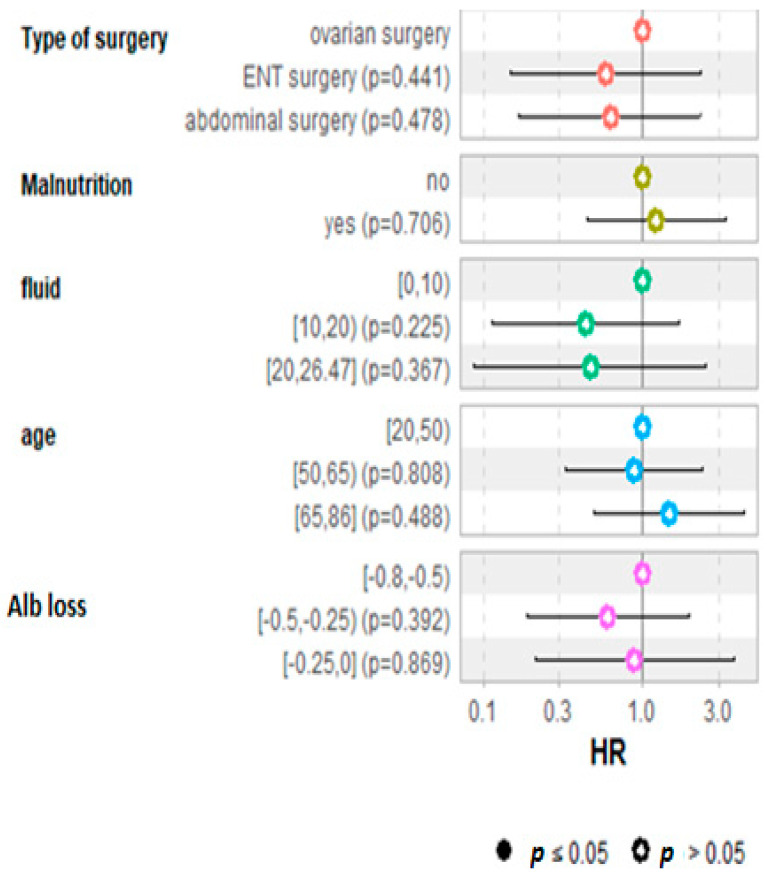
Cox regression hazard ratio for the survival at five years. No variable has any significant influence on survival.

**Table 1 ijerph-19-03394-t001:** Patients characteristics.

Characteristics	Ovarian Surgery N = 30	ENT Surgery N = 20	Abdominal Surgery N = 13	*p*-Value
Age	54 (37, 62)	62 (51, 70) $	55 (49, 61)	0.077
Weight (kg)	59 (52, 69)	66 (59, 80)	78 (74, 88) *	0.013
Height (cm)	161 (157, 168)	168 (160, 172) $	169 (161, 176) *	0.018
ASA score	-	-	-	0.8
2	25 (48.08%)	17 (32.69%)	10 (19.23%)	-
3	5 (45.45%)	3 (27.27%)	3 (27.27%)	-
Malnutrition	3 (25.00%)	8 (66.67%)	1 (8.33%)	0.017
Cardiovascular disease	-	-	-	0.920

ABD vs. DBK, *p* < 0.05; * ENT vs. DBK, $ *p* < 0.05.

**Table 2 ijerph-19-03394-t002:** Intraoperative characteristics with regard to type of surgery.

Characteristic	Ovarian Surgery N = 30	ENT Surgery N = 20	Abdominal Surgery N = 13	*p*-Value
Surgery duration (hour)	6.50 (5.00, 7.00)	7.00 (6.00, 7.00)	6.00 (5.00, 6.00)	0.2
Hemoglobin intraoperative variation %	−0.21 (−0.26, −0.08)	−0.17 (−0.24, −0.2)	−0.15 (−0.23, −0.14)	>0.962
Albumin intraoperative variation %	−0.49 (−0.52, −0.38)	−0.19 (−0.26, −0.12)	−0.31 (−0.48, −0.24)	<0.001
Intraoperative crystalloids (mL/kg/h)	10.9 (8.5, 14.0)	7.1 (6.3, 8.4)	9.5 (7.4, 12.8)	0.010
Perioperative colloids (mL/kg/h)	2.20 (1.54, 3.99)	0.00 (0.00, 0.95)	1.35 (0.79, 2.41)	<0.001
Perioperative amount of fluids (mL/kg/h)	14 (11, 17)	8 (0, 10)	8 (8, 15)	<0.001
Perioperative urinary output (mL/h)	72 (37, 79)	112 (80, 156)	72 (44, 101)	0.067

Values are mean (minimum–maximum).

**Table 3 ijerph-19-03394-t003:** Regression model for the maximum variation of albumin with regard to ASA score and DBK surgery.

Characteristics	Beta	95% CI	*p*-Value
ASA score			
2	-	-	-
3	−5.9	−15, 3.5	0.2
Type of surgery			
DBK	-	-	
ENT	26	17, 35	<0.001
ABD	15	5.4, 2.4	0.003
Intraoperative amount of fluid (mL/kg/h)	0.18	−0.42, 0.72	0.6
Intraoperative variation of hemoglobin	0.46	0.2, 0.72	0.001

## Data Availability

All data are available on demands.

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
