# Peer review of "Serum Albumin Kinetics in Major Ovarian, Gastrointestinal, and Cervico Facial Cancer Surgery"

_ijerph, 2022, doi:10.3390/ijerph19063394_

Round 1

Reviewer 1 Report

Motamed et al, investigated the kinetics of serum albumin levels in the patients  who underwent  surgical operation of ovarian , gastrointetinal and cervico facial cancers. Although serum albumin levels were reduced in all surgical operations, it was not co-related with the survival. There are no novelarities in this study. In addition, sample sizes are too small.

Author Response

Response to reviewer 1 comment 

Motamed et al, investigated the kinetics of serum albumin levels in the patients  who underwent  surgical operation of ovarian , gastrointetinal and cervico facial cancers. Although serum albumin levels were reduced in all surgical operations, it was not co-related with the survival. There are no novelarities in this study. In addition, sample sizes are too small.

We thank the reviewer for carefuly reading our manuscript 

The originality of this study  is related to the fact that we compared 3 types of  different cancer surgery , to our knowledge most of the studies in this field considered one type of surgery only and mostly major abdominal surgeries 

The above sentence has been inserted in the discussion section 

We agree that the sample size was too small with regard to long term outcome but the primary objective was to detect a the rate of the  decrease in serum albumin during surgery and the initial sample size was calculated for this purpose , unfortunately we had too many drop outs because of the decrease of  the predicted duration of surgery , mostly because the significant progression of cancer cancelling therefore major surgery. 

We rechecked  again very  recent litterature most of the papers exploring similar  subject have limited number of patients like our study 

Plasma Volume Expansion and Fluid Kinetics of 20% Albumin During General Anesthesia and Surgery Lasting for More Than 5 Hours
Michaela Gunnström, MD,*et al  2022 in  press (20 patients )

Electrolytes, albumin and acid base equilibrium during laparoscopic surgery Giovanni SABBATINI et al: Minerva Anestesiologica 2021 December;87(12):1300-8  (38 patients)

Indeed initially we had 73 patients ,  but for the final analysis only  data for 63 patients was adequately available considering exclusion criteria.

Reviewer 2 Report

In this manuscript, the authors aim to detect intraoperative albumin variations in three major cancer surgeries and to assess mid-and long-term survival and possible parameters affecting survival. Surgeries with a duration of more than five hours, along with other criteria, were considered for the current study. The manuscript is of sufficient importance; however, some issues need to be addressed as detailed below.

1-The authors selected patients requiring debulking (DBK) ovarian surgery as one study group. However, the authors did not clarify the type of debulking surgery; primary or interval. How was this parameter considered in the analysis if interval debulking surgery candidates were included (i.e, weeks after receiving neoadjuvant chemotherapy)?   

2-The primary goal of the study is to detect intraoperative albumin variations in three major cancer surgeries. However, another essential record has been overlooked: the albumin level at the time of diagnosis compared to the immediate pre-operative record and whether there are potential differences linked to the observed perioperative variations.

3-It is assumed that the DBK ovarian surgery only includes female patients. Some studies pointed to age and sex variation in serum albumin concentration and showed that values in females decrease more rapidly but become close to male values at age 60. The authors are encouraged to address this point because the DBK ovarian surgery group showed the highest variation in ALB level.

4-Concerning the debulking surgery group, were all successful in achieving cytoreduction? Further, were tumor grade and stage considered in the analysis, and whether they impact the perioperative ALB level?

5- Some sentences were confusing, indeed. As an example, Lines 57-59: Inclusion criteria: Patients requiring debulking (DBK) ovarian surgery, major abdominal surgery (ABD), or major cervicofacial surgery (ENT) with a duration of more than five hours were included in order to have four data points of assessment. While Lines 64-65 regarding exclusion criteria read “and actual surgery duration of less than six hours”. So, is five and half hours excluded or included? Furthermore, is it 4 points of assessment or 6 points according to Table 1?

6- Other points:

Line 4: cyrus ………Should C be capitalized?

Lines 127-128 (….Table 1 at the incision time through six postoperative hours). Line 131: Table 1. Perioperative blood samples. Is it postoperative or perioperative?

Line 24: a posteriori lack of potency?

The manuscript is likely to benefit from another round of editing.

Author Response

We thank the reviewver for carefully reading our manuscript we tried to satisfy all his suggestions 

1.The authors selected patients requiring debulking (DBK) ovarian surgery as one study group. However, the authors did not clarify the type of debulking surgery; primary or interval. How was this parameter considered in the analysis if interval debulking surgery candidates were included (i.e, weeks after receiving neoadjuvant chemotherapy)?  

Thank you for your suggestion in this selected group of patients we only considered all patients who had surgery with  an average of 3-4  weeks after a (4-6)  cycles of chemotherapy.  In addition all limits of excisions were free of disease in all three sub-groups of patients.

2-The primary goal of the study is to detect intraoperative albumin variations in three major cancer surgeries. However, another essential record has been overlooked: the albumin level at the time of diagnosis compared to the immediate pre-operative record and whether there are potential differences linked to the observed perioperative variations.

We agree with the reviewer that alb at the time of diagnosis is a pertinent paramater and was not explored but this omission was somehow on purpose since it has already been discussed and investigated however as asked by the reviewer  we included a sentence about this issue on  into the discussion section as a possible shorthcoming 

In addition, we did not consider the amount of serum albumin at the time of primary diagnosis of the disease since it was not the primary objective of the study nevertheless albumin is basically an additional tool for surgery risk stratification rather than a nutritional marker. 

3-It is assumed that the DBK ovarian surgery only includes female patients. Some studies pointed to age and sex variation in serum albumin concentration and showed that values in females decrease more rapidly but become close to male values at age 60. The authors are encouraged to address this point because the DBK ovarian surgery group showed the highest variation in ALB level.

Thank you for your suggestion , we adressed this point and add a sentence in the discussion section 

It should also be emphasized that in our group of ovarian cancer patients a gender distribution bias could be considered as a confounding factor (23)

4-Concerning the debulking surgery group, were all successful in achieving cytoreduction? Further, were tumor grade and stage considered in the analysis, and whether they impact the perioperative ALB level?

yes we provided the numbers  of optimal and suboptimal surgery ,in addition we rechecked again  details of surgery repports and all surgery were considered successful, however  one patient had incomplete surgery and was deleted from the final analysis.

in addition  giving that our cancer hospital is  specialized in cancer surgery and have agressive attitude in treatement of these tumors the difference between optimal and suboptimal debulking is very slim

Concerning possible impact of  different tumors on the perioperative level of albumin , we did not assessed this  issue since the main objective was to explore the rate of decrease of albumin whatever type of tumor

we specify  that survival outcome was not our primary objective therefore and our primary results was the fluctuation of serum albumin concentration 

5- Some sentences were confusing, indeed. As an example, Lines 57-59: Inclusion criteria: Patients requiring debulking (DBK) ovarian surgery, major abdominal surgery (ABD), or major cervicofacial surgery (ENT) with a duration of more than five hours were included in order to have four data points of assessment. While Lines 64-65 regarding exclusion criteria read “and actual surgery duration of less than six hours”. So, is five and half hours excluded or included?

We apologized for this mistake the correct time is 5 hours and we corrected it through the manuscript 

Furthermore, is it 4 points of assessment or 6 points according to Table 1?

Table 1 was removed and  datapoints assessements are now clearly inserted inside the flowchart . 

6- Other points:

Line 4: cyrus ………Should C be capitalized? yes thank you 

Lines 127-128 (….Table 1 at the incision time through six postoperative hours).

corrected 

Line 131: Table 1. Perioperative blood samples. Is it postoperative or perioperative?(it is intraoperative )

Line 24: a posteriori lack of potency? Thank you for your suggestion , we finally deleted this sentence.

Reviewer 3 Report

Serum albumin kinetics in major ovarian, gastrointestinal, and cervico facial cancer surgery and long-term outcomes by Motamed et al designed to detect albumin variations in three major cancer surgeries: ovarian debulking (DBK), major abdominal gastrointestinal surgery (ABD), and major cervico facial, or ear, nose and throat cancer surgery (ENT) is an interesting work.

 A few concerns need to e addressed

  1. As a general reader, the relevance of the work is not clear  in the introduction
  2. Too many tables. Put only important tables in the main content. It is good to represent as bar diagrams
  3. Figure legends are not clear
  4. Clarity should be there in the methodology adopted
  5. Discussion and conclusion need to be expanded
  6. Suggest having any flow chart for representing the methodology adopted.
  7. Have you consulted with statistician for choosing the scale of analysis
  8. More details of the ethica approval including the number and institution. Should give more importance

Author Response

1)

  1. As a general reader, the relevance of the work is not clear  in the introduction

Thank you for carefuly reading the manuscript and pointing constructively to issues 

We modified sentences in relation to primary and secondary objectives of the study , basically this study focused on intraoperative kinetics of serum albumin in 3 majors cancer surgery patients , ovarian , gastrointestinal and cervica facial (ENT) patients which has not been studied previously ,

This issue  was important to us since pretreatemnt albumin infusion is reported  to affect survival in some studies , therefore the extent of the  predictive decrease in surgery of serum albumin  in different situations   could affect some protocols (for  ex if in some surgeries there is a predicted significant decrease in albumin , therefore an additional dose of pretreatment could be used 

regarding the secondary objective , we wanted to assess if a patient with higher loss of albumin during surgery would have a lower survival rate (unfortunately this could not be demonstrated basically because of a sample size issue   

  1. Too many tables. Put only important tables in the main content. It is good to represent as bar diagram

We deleted two tables  (table 1 and table 5), we cannot see which data  should be present in bar diagramms , if the reviewer have specific request we are ready to perform it 

    • Figure legends are not clear , The figure legends have been updated we hope they are easier to understand , 
    • Clarity should be there in the methodology adopted 
    • We adjusted  the method section to make it more understandable 
    • Discussion and conclusion need to be expanded As requested we expanded the discussion and conclusion 
    • Suggest having any flow chart for representing the methodology adopted. A new flowchart has been added 
    • Have you consulted with statistician for choosing the scale of analysis?
    • Yes absolutely Dr Weil has a master  Degree of publichealth and biostatistics, and he also consulted our hopsital statistician at the time of the study  
    • More details of the ethical approval including the number and institution. Should give more importance 
    • Yes the number of the ethical commette approval is given in the method section the patient informed consent is available in french if someone  want to consult it we provided the ethical committei approval and the blank informed consent  to the editors

Round 2

Reviewer 1 Report

I understood the author's claims. However, because the results of this study are predictable, it was necessary to analyze additional parameters. Further studies are desirable in the future.  It is better to change one miner point; `long-term outcome'  in title is not proper, since significant results were never obtained.

Author Response

Dear reviewer 

Thank you for your suggestion 

We deleted long term outcome from the title 

and we also changed in the secondary objective from long term mortality to  lon term follow up